# Diversification rate vs. diversification density: Decoupled consequences of plant height for diversification of Alooideae in time and space

**Florian C. Boucher** [1,2] *, **Anne-Sophie Quatela** [3], **Allan G. Ellis** [1], **G. Anthony Verboom** [4]

**1** Department of Botany and Zoology, University of Stellenbosch, Matieland, South Africa, **2** Univ. Grenoble Alpes, Univ. Savoie Mont Blanc, CNRS, LECA, Grenoble, France, **3** Department of Biological and Environmental Sciences, University of Gothenburg, Göteborg, Sweden, **4** Department of Biological Sciences, University of Cape Town, Rondebosch, South Africa

* fcboucher@univ-grenoble-alpes.fr

**Data Availability Statement:** In an effort to make all of our analyses fully reproducible, a script of all R-based analyses as well as the datasets necessary

## Abstract

While biodiversity hotspots are typically identified on the basis of species number per unit area, their exceptional richness is often attributed, either implicitly or explicitly, to high diversification rates. High species concentrations, however, need not reflect rapid diversification, with the diversity of some hotspots accumulating at modest rates over long timespans. Here we explore the relationship between diversification in time vs. diversification in space and develop the concept of *diversification density* to describe the spatial scale of species accumulation in a clade. We investigate how plant height is associated with both aspects of diversification in Alooideae, a large plant subfamily with its center of diversity in the Greater Cape Floristic Region. We first reconstruct a time-calibrated phylogeny for Alooideae and demonstrate an evolutionary tendency towards reduced plant height. While plant height does not correlate with diversification rate across Alooideae it does so with diversification per unit space: clades of small plants tend to have the highest diversification densities. Furthermore, we find that diversification in time vs. space are uncorrelated. Our results show that diversification rate and density can be decoupled, and suggest that while some biodiversity hotspots might have been generated by high diversification rates, others are the product of high diversification density.

## Introduction

The spatial distribution of biodiversity is extremely uneven [1–3], with some regions of Earth– biodiversity hotspots–harboring exceptionally high levels of biodiversity per unit area [4] and other areas being relatively species-poor. Although diversity patterns may be studied from both an ecological and evolutionary perspective, with the two approaches providing complementary insights [5,6], understanding the origins and origination of diversity is a fundamentally evolutionary challenge that has been most commonly addressed in a clade-centric context. Consistent with the observation that accelerated diversification accounts for the high species richness of some mountainous [7,8] and island [9,10] systems, and of the

 

to run them are available at the following address: https://github.com/fcboucher/Alooids-height.

**Funding:** F.C.B. acknowledges support by a grant from the Claude Leon Foundation.

**Competing interests:** The authors have declared that no competing interests exist.

Mediterranean Basin [11], evolutionary explanations for the origins of biodiversity hotspots have traditionally emphasized the temporal component of diversification (*i.e.* elevated rates of speciation or lowered rates of extinction per unit time) [12–14]. In some regions, however, such as the Cape Floristic Region of southern Africa and seasonally-dry neotropical forests, plants have achieved high species richness despite a moderate pace of diversification, because species have been able to accumulate over long time periods [15,16]. This suggests that elevated diversification rate is not a general feature of biodiversity hotspots and that there is a need to consider additional explanations that are more explicitly linked to elevated accumulation of species in a given area.

In this article we explore the parallels and differences between diversification in time and diversification in space. In order to avoid confusion between these two aspects of diversification, we introduce the term *diversification density* for diversification per unit distance in contrast to *diversification rate*, which is traditionally used for diversification per unit time. Although it has never been formalized as we propose here, the role of space in clade diversification has been widely discussed in the macroevolutionary literature. A clade's range size, for example, is known to be a key determinant of its diversity [17], with the colonization of a new region often triggering rapid evolutionary radiation thanks to the availability of new ecological opportunity [18]. But smoother expansion of a clade's range should also lead to a higher species richness thanks to an increase in the environment's carrying capacity, either in purely spatial terms [19,20] or in terms of environmental heterogeneity [20,21]. In a related corpus of ideas, the theory of island biogeography explains the species-area relationship largely as an outcome of the positive effect of island area on immigration (which frequently leads to anagenetic speciation) and its negative effect on extinction [22,23]. Extensions of this theory even show that *in situ* speciation on islands correlates positively with island area [24]. Finally, global analyses find that the area occupied by a flowering plant family is by far the best predictor of its species richness [25] and a detailed study of the Cyperaceae suggests that high richness in a clade is most readily attained by fine partitioning of species' ranges [26].

Diversification density and rate are potentially linked since both measures relate to the accumulation of species in a clade, in space and time respectively. Both empirical and theoretical studies have shown that the probability of speciation increases with the area occupied by an ancestral species [23,27]. This is partly because larger areas usually feature a greater diversity of ecological conditions and thus provide more opportunities for ecological speciation, but also because larger areas enable greater isolation by distance and often harbor more potential barriers to gene flow, thereby increasing the probability of purely geographic speciation. Extinction provides an even more natural connection between space and time, since extinction actually coincides with the moment a species' range reaches an area of zero. Range size is consequently one of the strongest correlates of extinction risk [28]. Thus, all else being equal, the range size of an ancestral species should correlate positively with speciation rate and negatively with extinction rate. As a result, diversification density and diversification rate should generally be positively correlated.

Diversification density and rate might, however, be decoupled under some conditions that depend on a clade's ecology and historical biogeography. For example, one could envision a scenario in which a clade partitions its environment finely through increased ecological specialization and/or reduced dispersal distances, thus leading to a high level of species packing but not necessarily a high diversification rate. The initial phase of colonization of a new area might also lead to an increase in diversification rate together with an even faster expansion of the clade's range (*i.e.* a lower diversification density overall). Situations in which a given region is colonized several times by members of the same clade should also lead to a decoupling of diversification rate and diversification density (e.g. [29]).

Arguments have recently been presented for why size (and in particular height) should influence diversification in plants, suggesting that small plants should generally enjoy both higher rates of speciation and lower rates of extinction than larger ones [30]. While these arguments were presented in the context of diversification per unit time, they apply at least as well, and probably even better, to diversification per unit area. Small plants should have higher speciation densities because they have, on average, shorter dispersal distances than larger plants [31], which makes geographic isolation of populations easier over smaller spatial scales and thus favors divergence under any speciation model [32,33]. In addition, because of their size, small plants can perceive much finer environmental heterogeneity [34,35] and generally grow at higher population densities than tall plants [36]. This should lead to both stronger and more efficient divergent selection for local adaptation across short environmental gradients, thus favoring divergence under any model of ecological speciation [37,38]. Reduced extinction in small plants should arise as a consequence of both increased population size [36], which provides a general buffer against extinction, and a greater propensity for niche separation on heterogeneous resource patches [35], thus limiting competition at high species densities and so reducing extinction risk.

We investigate the evolution of plant height and its consequences on diversification rate (per unit time) and diversification density (per unit distance) in Asphodelaceae subfamily Alooideae. Alooideae provides a good system for addressing such questions as it is a large group of c. 500 species [39] which spans three orders of magnitude in height, from miniature species in the genus *Haworthia* that may grow just a few centimeters tall, to trees of the genus *Aloidendron* which can reach heights of more than 15 m (Fig 1). Using a new phylogenetic hypothesis for Alooideae and state-of-the-art comparative methods, we first investigate the evolution of plant height across Alooideae and uncover a tendency for Alooideae to evolve towards shorter stature. We then investigate the two aspects of diversification and find that diversification density and rate are decoupled across Alooideae. Contrary to our predictions, our results show that lineages of short Alooideae do not diversify faster than tall ones. However, we find strong evidence that diversification density correlates with plant height, being highest in the shortest Alooideae.

## Material and methods

Unless stated otherwise, all analyses were run in the R statistical environment [40]. In an effort to make all of our analyses fully reproducible, a script of all R-based analyses as well as the datasets necessary to run them are available at the following address: https://github.com/fcboucher/Alooids-height.

### Study group

Alooideae is the largest subfamily in the Asphodelaceae family, containing about 500 species [39]. All of these species occur in Africa and adjacent regions, with centers of diversity in the Arabian Peninsula, Madagascar, and southern Africa [41]. Eleven Alooideae genera are currently recognized, the most iconic of which, *Aloe* L., has recently been re-circumscribed [39,42]. This has led to the creation or reinstatement of the genera *Aloidendron* (A. Berger) Klopper & Gideon F. Sm., *Aloiampelos* Klopper & Gideon F. Sm., *Aristaloe* Boatwr. & J. C. Manning, *Gonialoe* (Baker) Boatwr. & J. C. Manning, and *Kumara* Medik. Similar taxonomic changes have led to re-circumscription of the genus *Haworthia* Duval and the creation of the genus *Haworthiopsis* G. D. Rowley. Finally, Alooideae also contains the genera *Tulista* Raf., *Gasteria* Duval and *Astroloba* Uitewaal. In the rest of this article we follow the latest taxonomic revision of the group [39]. Of all the Alooideae genera, only *Aloe* extends across Africa, the

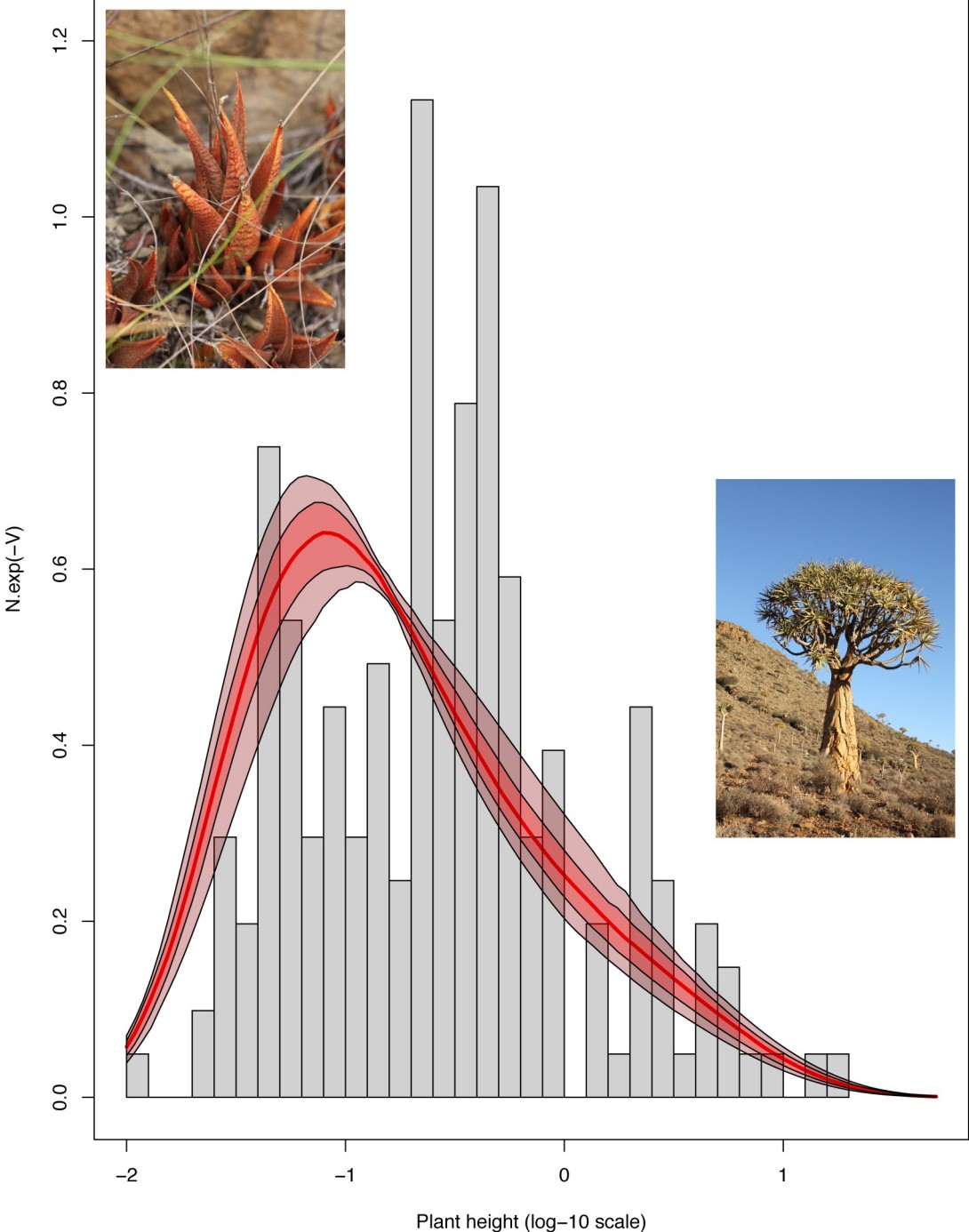

**Fig 1. Current distribution and evolution of maximum vegetative height across Alooideae.** The observed distribution of height across current Alooideae is shown as a histogram. Macroevolutionary landscapes estimated under the BBMV4 model over the phylogenetic posterior have been superimposed: the solid line shows the median landscape estimated over the 100 posterior trees. Polygons around the median of the phylogenetic posterior show the 50% (dark red) and 90% (light red) quantiles of macroevolutionary landscapes. Inset pictures show representatives of the shortest and tallest genera of Alooideae: *Haworthiopsis venosa* (10 cm) on the left, and *Aloidendron dichotomum* (6 m) on the right. Both pictures by F. C. Boucher.

Arabian Peninsula, and Madagascar, the remaining ten genera being restricted to southern Africa (*i.e.*, Botswana, Lesotho, southern Mozambique, Namibia, Lesotho, South Africa, Swaziland, Zimbabwe). Alooideae is an emblematic group of succulent plants: all species exhibit some form of leaf succulence and, as far as is known, they all possess CAM photosynthesis [41]. While most Alooideae have long-tubed flowers that are bird pollinated, species of *Haworthia* as well as some species of *Aloe* seem to be exclusively pollinated by insects [43,44]. Fruits are dehiscent, with seed dispersal being either passive or by wind, and plant height being shown to influence dispersal distance positively in some species [45–47].

## Phylogenetic inference and dating

While several phylogenetic hypotheses for Alooideae exist [39,41,48], the utility of these trees for our purposes was limited primarily by sampling in each instance being focused on a particular genus. We thus inferred a new dated phylogeny for Alooideae as a whole combining all sequence data available on public databases. We assembled a dataset of six chloroplast markers (matK, rbcL, trnH-psbA, trnL intron, trnl-F internal spacer, rsp16) and one nuclear marker (ITS1) for 204 species of Alooideae. Nucleotide sequences originating from previous studies [39,41,48] were obtained from GenBank (ncbi.nlm.nih.gov/genbank/) and Phylota (phylota. net/). Accession numbers for all sequences used can be found in S1 Table in S1 Appendix. In total, the available data accounted for about 40% of Alooideae species, and were distributed as follows: 30% of *Aloe* species (118 spp. sampled), 100% of *Aloidendron* (6 spp.), 78% of *Gasteria* (18 spp.), 86% of *Haworthia* (33 spp.), 100% of *Tulista* (4 spp.), 56% of *Haworthiopsis* (10 spp.), 71% of *Aloiampelos* (5 spp.), 100% of *Astroloba* (6 spp.), 100% of *Kumara* (2 sp.), 33% of *Gonialoe* (1 sp.) and 100% of *Aristaloe* (1sp.). Six outgroups from the family Asphodelaceae were added to this dataset, namely *Asphodelus aestivus* Brot., *Asphodeline lutea* (L.) Rchb., *Bulbine semibarbata* (R.Br.) Haw., *Bulbine wisei* L.I.Hall, *Bulbine succulenta* Compton, and *Jodrellia macrocarpa* Baijnath. Automatic sequence alignment was performed using MUSCLE [49] and manually adjusted. The different markers were aligned and adjusted separately and were then concatenated into a single DNA matrix of 6,154 bp containing 55% missing data. Phylogenetic inference was conducted in a Bayesian framework using MrBayes [50], with two partitions: ITS versus all chloroplast markers concatenated. We used the best substitution model for each partition, as determined using jModeltest v. 2.1.10 [51]: the GTR+Γ model for chloroplast markers and the HKY+Γ model for ITS. The phylogenetic analysis then consisted of three independent runs with four chains each that were run for 20 million steps. After convergence of these runs had been verified, we build a maximum clade credibility (hereafter 'MCC') tree from the posterior of the analysis, but also randomly selected 100 trees from the posterior (hereafter 'posterior trees') so that phylogenetic uncertainty could be incorporated in subsequent analyses.

Since no fossils are available for Alooideae, we dated the phylogenies (both the MCC and the posterior trees) using secondary calibration, following the procedure used in a previous phylogenetic study focused on the genus *Aloe* [41]. To do so we used penalized likelihood [52] as implemented in the *chronos* function of the R package *ape* [53]. For each tree we compared six different values of the smoothing parameter λ (0.01,0.1,0.5,1,2,10), of which we retained only the one that gave the lowest ΦIC [54]. We calibrated the crown age of Asphodelaceae using a fixed age of 34.2 Ma (95% CI: 24.0–46.9 Ma), which is the median age obtained in a phylogenetic study of Asparagales [55]. While secondary calibration is generally prone to error propagation [56], this is of lesser concern here since we were principally interested in relative diversification rates within Alooideae (see below).

## Evolution of plant height

Maximum vegetative height could be retrieved for all species included in our phylogeny but one, (i.e. 203 species in total) using information from taxonomic accounts [44,57,58]. The distribution of plant height among Alooideae (log$_{10}$ transformed in all subsequent analyses) was positively skewed (skew = 0.44, D'Agostino test $p$ = 0.01, Fig 1).

General patterns of height evolution across Alooideae were inferred using the Fokker-Planck-Kolmogorov (hereafter, FPK) model, which is the most general model for the evolution of continuous characters [59]. In the FPK model, characters evolve along a phylogeny under both random diffusion and deterministic forces of any possible shape and strength. Deterministic forces are represented by a function $V(x)$ describing the evolutionary potential associated with each value of the trait, $x$: traits are attracted towards regions of lowest evolutionary potential. This potential in turn determines the distribution of the trait at equilibrium, which can be interpreted as a macroevolutionary landscape, i.e. a surface depicting trait values that have been favored over the course of evolution [60]. The FPK model admits classic models of trait evolution as special cases. Brownian motion (BM) [61] corresponds to a case in which there is only diffusion but no deterministic forces acting on trait evolution (i.e., the macroevolutionary landscape is flat). The Ornstein-Ulhenbeck model (OU) [62], which is often used to describe evolution towards a single, symmetric, trait peak, is also a special case of the FPK model with a Gaussian-shaped macroevolutionary landscape. Furthermore, the FPK model can also accommodate reflective bounds on the evolution of quantitative characters, this particular version being termed the BBMV model. We fitted seven different models to the evolution of plant height on the phylogeny of Alooideae. The first of these did not incorporate bounds, being the BM model, the OU model and the FPK model with $V(x) = a.x^4 + b.x^2 + c.x$. However, we also fitted four different forms of the BBMV model, with $V(x) = a.x^4 + b.x^2 + c.x$ (BBMV4), $V(x) = b.x^2 + c.x$ (BBMV2), $V(x) = c.x$ (BBMV1) and $V(x) = 0$, which is the bounded Brownian motion model [63]. When fitting BBMV models, we fixed a lower bound at 1 cm for maximum vegetative height, which corresponds to the minimum height observed across Alooideae, and an upper bound of 50 m, which is much higher than the maximum height observed in Alooideae. BM and OU models were fitted using the *geiger* package [64], while FPK and BBMV models were fitted using the *BBMV* package [65]. Both packages provide directly comparable likelihood functions. Relative model fits were compared on the MCC tree using the Akaike Information Criterion (AIC), but parameters of the best fitting model were then estimated on 100 posterior trees.

## Influence of plant height on diversification rate

We tested the hypothesis that short plants diversify faster than tall ones [30] across Alooideae. This was done using STRAPP, a semi-parametric randomization method that tests for correlations between a trait and diversification rates [66]. To do this, we first fitted the BAMM diversification model [67] to the phylogeny of Alooideae, using default priors and an expected number of diversification rate shifts of one, as recommended for trees with less than 500 tips [68]. Two MCMC chains of $2 \times 10^6$ steps each were run, allowing for diversification rates to vary exponentially through time and accounting for the number of missing species in each genus separately. We then ran the STRAPP test using posterior diversification rates obtained from the BAMM analysis and log$_{10}$-transformed plant height measurements for species at the tips of the phylogeny, using 999 randomizations. We preferred to analyze net diversification rates rather than speciation and extinction rates separately, given that these components might be difficult to disentangle [69].

In order to compare directly the influence of plant height on diversification rate and density (see below), we complemented the STRAPP analysis with a heuristic method. We measured the correlation between the net diversification rate measured in different clades and the median plant height in these clades. Net diversification rates were calculated as log(n)/t [70], where n is the number of species in a clade and t is its stem age. Rather than measuring this relationship across the 11 genera of Alooideae, we used unnamed clades of the Alooideae phylogeny that were cut at a given time-point (*i.e.* clades of at least two species descending from each lineage present in the phylogeny at this time point) and used as data points for this analysis. Our choice to measure net diversification rates using stem rather than crown ages comes from this tree-slicing procedure. The procedure was repeated for time-cutoffs spaced 1 Myr apart, from 5 Ma to 15 Ma (outside of these values the unnamed clades delimited were too small or too few, respectively), with the analyses at each cutoff being repeated over the 100 posterior trees. For these analyses, the backbone phylogeny connecting clades was used in order to run phylogenetic regressions, with a λ model [71] for the evolution of the residuals. Note that this second analysis does not account for missing species in our phylogeny of Alooideae since it uses unnamed clades for which exact species numbers are unknown.

## Influence of plant height on diversification density

We then tested whether clades of short plants tend to accumulate more species per unit area than clades of tall plants, *i.e.* whether plant height correlates negatively with diversification density. This was done by fitting phylogenetic regressions testing for an influence of median plant height on diversification density among clades. Here again we used unnamed clades of the Alooideae phylogeny cut at given time-points ranging from 5 Ma to 15 Ma, and lineages with a single species surviving to the present were excluded from the analysis. For each cutoff, the relationship between diversification density and plant height was measured over each of the 100 posterior trees, using phylogenetic regression.

As introduced above, and subject to certain assumptions (see Discussion), diversification density should quantify the spatial scale at which lineages diversify. Although it is widely recognized that clades differ with respect to levels of species packing, no standardized measure of diversification density has yet been developed. Here we present a phenomenological measure of net diversification density which, like the metric for net diversification rate ($log(n)/t$), is neutral with respect to the processes that underpin between-lineage variation in diversification density. Our measure of net diversification density is calculated as $log(n)/sqrt(A)$, where *n* is the total number of species in the clade and *A* the area occupied by the clade. In this metric, as in the standard measure of net diversification rate, the numerator is log-transformed to accommodate the exponential accumulation of species in phylogenetic trees and so ensure that net diversification density is expressed on a per-lineage basis. The denominator, on the other hand, is square root-transformed since (in terrestrial organisms) diversification and subsequent range movement are possible along two spatial dimensions. The final metric thus obtained is directly comparable to the metric for diversification rate: where the latter quantifies the mean waiting time to accumulation of an additional species (*i.e.*, speciation minus extinction), our diversification density metric quantifies the mean straight-line distance over which additional species are accumulated. Assuming that *A* is expressed in $km^2$, net diversification density, as outlined above, is expressed in units of $spp.km^{-1}$. Simulations of individuals migrating and diversifying in a spatially explicit context reveal that this metric correlates strongly with several processes that we expect to influence the spatial scale of diversification. This is not true for three other possible metrics (*i.e.*, $n/A$, $log(n)/A$, and $log(n)/log(A)$) that we tested (S2 Appendix). Specifically, our preferred metric correlates negatively with both the migration

rate of individuals and the dimensions of the landscape (*i.e.* slow migration and/or small area available to a clade promote higher diversification densities), and positively with the carrying capacity of local communities (*i.e.* large populations achieve higher diversification densities). Crucially, too, it is insensitive to speciation mode (S2 Appendix).

In order to measure the area of the distribution of different clades, we used occurrence records from the global biodiversity information facility (www.gbif.org, hereafter GBIF), downloaded on 2019/12/13. Since GBIF is known for its many errors [72], we carefully curated these records, As a first step, all points outside of the natural distribution of Alooideae (*i.e.* continental Africa, Madagascar, and the Arabian peninsula) were excluded. We then removed all records from botanical gardens, based on searches of the strings 'botanical' and 'garden' in the description field of the GBIF records. Finally, we produced maps of all remaining occurrence records for each species and manually checked them against known distributions (especially from [48]). With these cleaned occurrence records at hand, we measured the area of the distribution of various clades as the area of the convex hull formed by all known occurrence points for species of the clade.

## Diversification in time vs. space

Finally, we measured how both aspects of diversification, rate and density, are coupled across Alooideae. This was done by measuring the correlation between both measures across the eleven time cutoffs used to define unnamed clades and the 100 trees from the phylogenetic posterior.

## Results

### Phylogeny of Alooideae

We first verified that all three MCMC runs had reached convergence before combining them, discarding the first 10% of samples in each as burnin. In general our results confirmed the findings of previous studies that provided the original genetic data that we used but high uncertainty remained regarding phylogenetic relationships within Alooideae, both regarding relationships between and within genera (Fig 2). The genera *Aloe*, *Aloiampelos*, *Aloidendron*, *Astroloba*, *Gasteria* and *Tulista* were strongly supported as monophyletic (posterior probability, PP > 0.96), while the monophyly of *Haworthia* received lower support (PP = 0.67) (S1 Fig in S1 Appendix). The monophyly of *Gonialoe* could not be assessed since we only included one species of this genus, but its phylogenetic placement confirmed previous evidence. On the contrary, we did not infer the two species of *Kumara* as sister species, our analysis identifying them as the two earliest diverging lineages within Alooideae, contrary to previous findings [39,41,48]. Finally, we inferred the genus *Haworthiopsis* to be paraphyletic, confirming former findings [48,73]. Divergence time estimation on the MCC tree yielded a crown age of 29.1 Myr for Alooideae. There was wide variation in the crown ages estimated for different genera, the oldest being *Aloe* (20.8 Myr) and the youngest being *Astroloba* and *Tulista* (6.8 Myr for both).

### Evolution of plant height

Of the seven models of plant height evolution in Alooideae, an FPK model with a lower bound on plant height and the most complex form of the macroevolutionary landscape (BBMV4) fitted best (AIC$_w$ = 0.703, S2 Table in S2 Appendix). Most of the remaining AIC weight was shared between another bounded model with a simpler form of the potential (BBMV2, AIC$_w$ = 0.182) and an FPK model (AIC$_w$ = 0.089), while the OU model received very low support (AIC$_w$ = 0.025) and BM virtually none (AIC$_w$ = 1.03E-6). When fitting the best model to 100

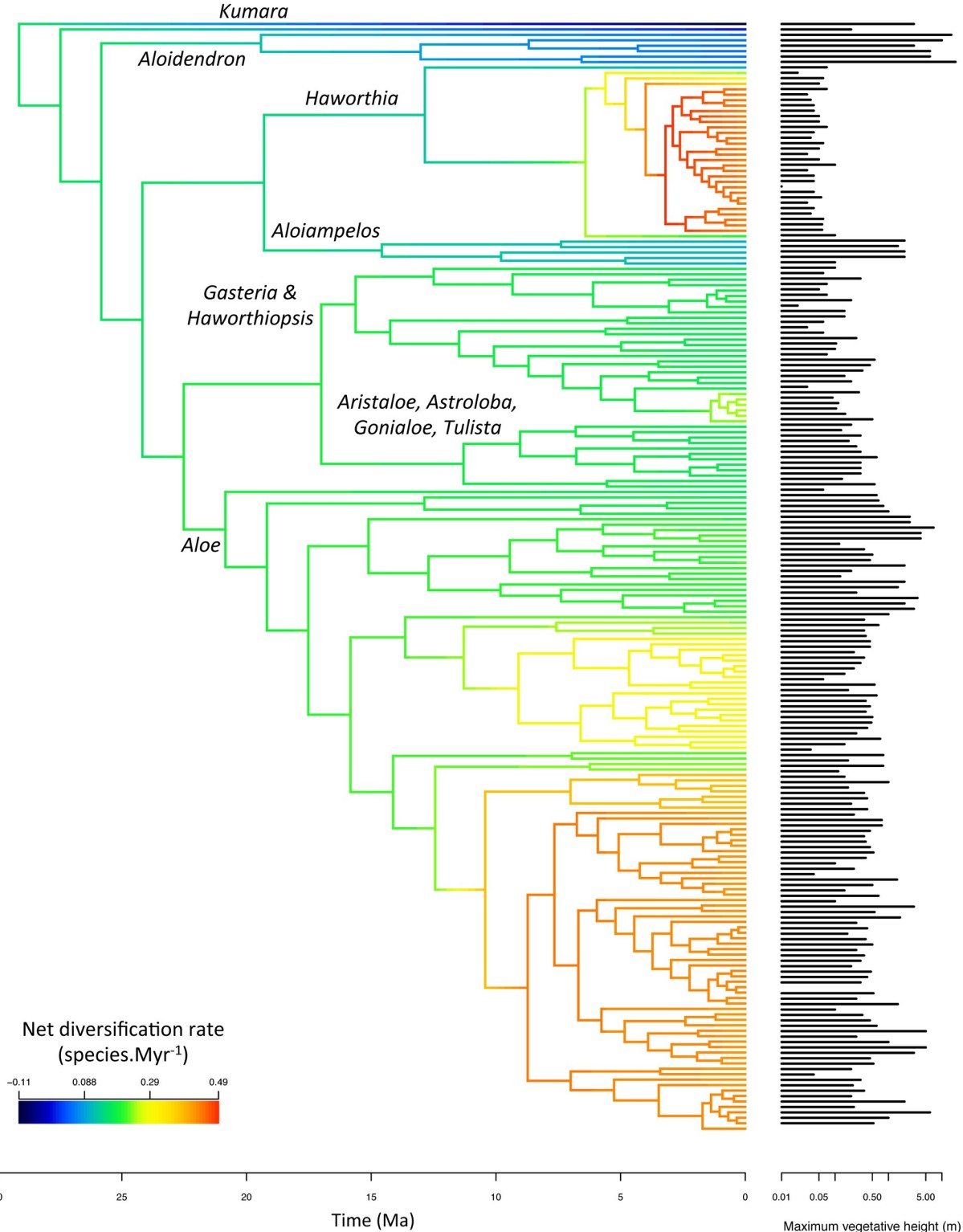

**Fig 2. Phylogeny, diversification and height in Alooideae.** The figure shows the maximum clade credibility tree of Alooideae. In order to illustrate the heterogeneity of diversification rates within Alooideae, branches in the tree have been colored according to their instantaneous net diversification rate, inferred using the BAMM software (Rabosky et al. 2013) under default settings and summarized across the posterior of the analysis. Bars on the right side of the phylogeny show the maximum vegetative height of each species, plotted on a $\log_{10}$-scale.

posterior trees, we found a clear tendency for evolution towards rather small heights, with a maximum of the macroevolutionary landscape estimated around a height of 8 cm (Fig 1). This value is lower than the median height of modern Alooideae, which is 28 cm. The MRCA of Alooideae was also estimated to have been taller than 8 cm (mean ML estimate across the 100 trees: 46.3 m, but 90% probability density interval averaged over the 100 trees: 0.13–50.0 m).

## Diversification in time vs. space

Both diversification rate and diversification density varied widely in Alooideae: the BAMM model suggested that net diversification rate ranged from 0.056 to 0.87 sp.Myr-1 (Fig 2) while diversification density varied from 2.2e-3 to 4.7e-2 sp.km$^{-1}$. The correlation between diversification rate and diversification density, measured on unnamed clades of the phylogeny was generally positive but only slightly so ($r = 0.24 \pm 0.12$ across all time cutoffs and posterior trees), and most importantly this relationship was non-significant at a 5% error rate in 94.4% of all comparisons (Fig 3).

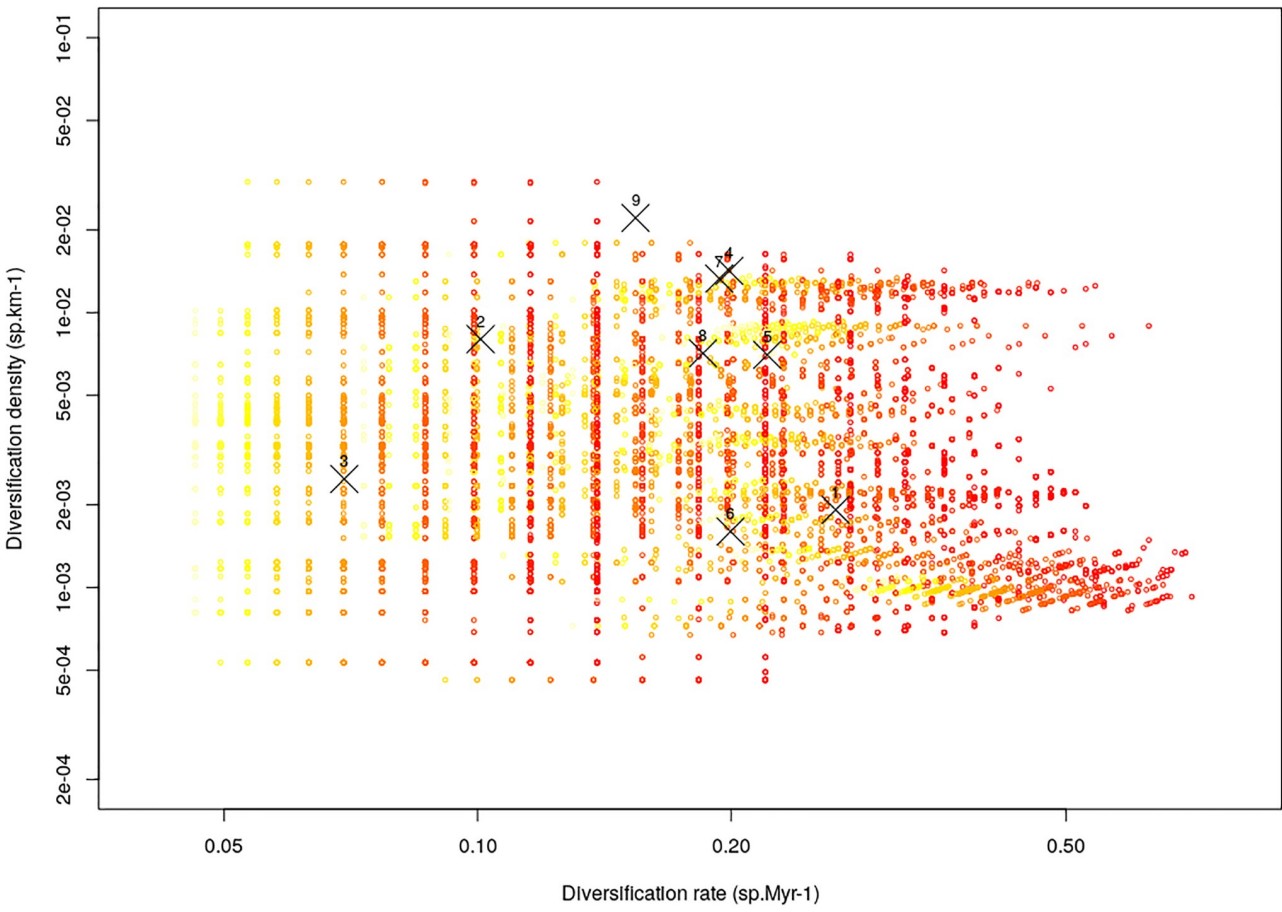

**Fig 3. Decoupling of two aspects of diversification.** The graph shows the relationship between the average diversification rate, *log(richness)/age*, and the average diversification density, *log(richness)/sqrt(area)* within clades. Black crosses show data for the nine genera of Alooideae for which this metric could be calculated, with numbers depicting each genus. 1: *Aloe*, 2: *Aloiampelos*, 3: *Aloidendron*, 4: *Astroloba*, 5: *Gasteria*, 6: *Gonialoe*, 7: *Haworthia*, 8: *Haworthiopsis*, 9: *Tulista*. Open circles present data for the 100 posterior trees and for all 11 time cut-offs, with the color scale ranging from 15 Myr old clades in yellow to 5 Myr old clades in red. In 98.7% of all comparisons, diversification rate and density were not significantly correlated.

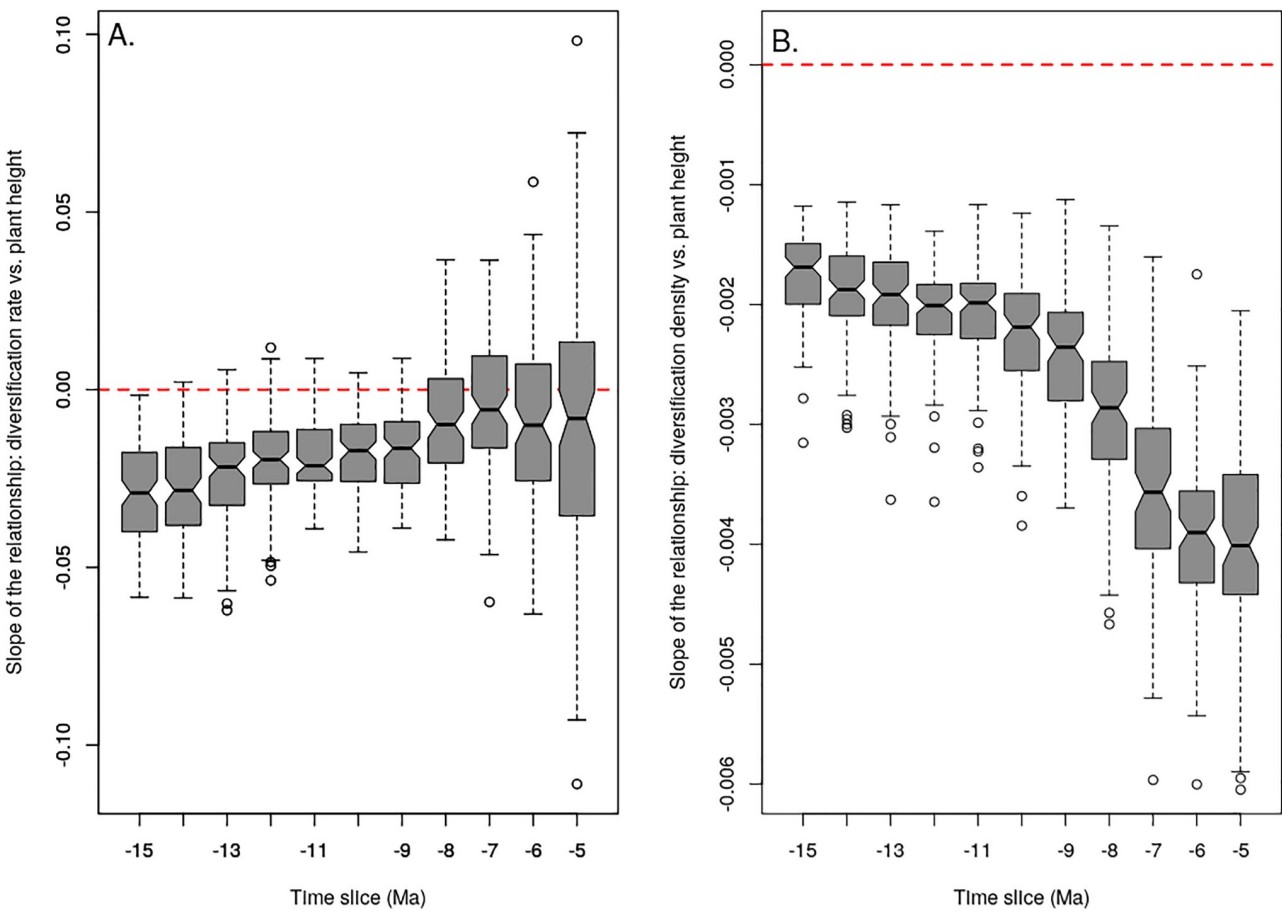

**Fig 4. Influence of plant height on diversification per unit time and per unit area.** In each panel, boxplots show the distribution of the slope of the relationship between one measure of diversification and median plant height in clades over the 100 posterior trees. Results for different time cut-offs are shown separately. The scale of the Y-axis varies between plots but both of them include the null slope, indicated by the dashed red line. A. Relationship between diversification rate and plant height. B. Relationship between diversification density and plant height.

### Influence of plant height on diversification rate and density

The 95% credible set of diversification rate shift configurations inferred using BAMM included 364 distinct configurations, but overall two main events were consistently detected: diversification rate increases within the genera *Haworthia* and *Aloe* (Fig 2). Contrary to our predictions, the STRAPP test indicated that plant height was not correlated with net diversification rate ($r$ = -0.13, p = 0.69) across Alooideae. There was evidence for a negative correlation between net diversification rate and median height in unnamed clades of the Alooideae phylogeny, but 95% confidence intervals for this correlation obtained across the phylogenetic posterior always included zero, except for the oldest cutoffs (Fig 4A). Statistical power varied slightly between time cutoffs since the number of unnamed clades ranged from 16 (median number over the 100 posterior trees cut at 15 Ma) to 30 (median number at 6 Ma). In contrast to diversification rate, we found a consistently negative relationship between plant height and diversity density across unnamed clades of the 100 posterior trees (Fig 4B). Results varied slightly when using different time cutoffs to delimit clades, but the relationship was significantly negative for all cutoffs (Fig 4B).

## Discussion

### A trend towards decreasing height in Alooideae

Maximum vegetative height varies over more than three orders of magnitude across modern Alooideae and its distribution is strongly skewed toward small sizes (Fig 1). By fitting a new class of evolutionary models for continuous characters that allows for both directional trends and bounds on trait values [59], we found that the most likely reason for the preponderance of short species among Alooideae is a tendency for evolution towards smaller plant stature. The optimum height estimated by the BBMV4 model is 8 cm, which, together with the inference of a lower bound on the height of Alooideae, explains the skewed height distribution across the group: most species evolve towards shorter heights, but since there is indeed a minimum bound, small stature species accumulate close to this bound. Importantly, since the MRCA of Alooideae is estimated to have been taller than 8 cm (90% probability density averaged over the 100 trees: 13 cm—50 m, average ML estimate: 46.3 m), there is good evidence for an actual reduction in plant height throughout the history of the group.

Reasons for the general evolutionary tendency towards smaller plant size are difficult to pinpoint. One might have expected that short plants are favored in the driest and least vegetated environments. Such low-productivity environments might not support the rates of carbon storage needed to produce long stems, and the reduced plant cover in such environments might in any case lead to minimal or no competition for light [74,75]. In addition, tall plants and in particular tall trees suffer a higher risk of xylem hydraulic failure from drought-stress [76]. Finally, small size may confer thermoregulatory benefits in summer-arid environments by enabling plants to inhabit shaded crevices near ground level [77]. We tested this hypothesis but did not detect any significant association between height and macro-environmental (*i.e.* climatic) conditions across Alooideae (results not shown). Another possibility is that micro-environmental conditions exert stronger selective pressure on plant height than the macro-environment. Many Alooideae species grow in rocky outcrops or even rock crevices, possibly as a means to escape competition from faster growing plants, to escape herbivory, or to escape fires to which succulents are generally vulnerable [78]. The poorly developed soils in rocky habitats likely select for smaller plants and in addition small plants are able to exploit shaded microsites in such environments, which are beneficial under a wide range of macro-environmental conditions found in southern Africa. Both factors could thus be responsible for the trend of decreased height that we inferred across the history of Alooideae.

Importantly, we show that the preponderance of short species in Alooideae is a result of this tendency for species to evolve towards smaller heights rather than being a product of accelerated diversification in lineages containing the shortest plants [30]. This is an important result which echoes long-standing evolutionary questions regarding the relative roles of cladogenetic vs. anagenetic evolution in creating skewed body size distributions across clades [79–82].

### Little evidence for higher diversification rates in short plants

While the two methods that we used to test for an influence of plant height on diversification rates gave slightly different results, they were consistent in showing no strong evidence for increasing diversification rates with decreasing plant height across Alooideae as a whole. This was contrary to our expectations, but theoretical arguments for why small plants should generally diversify faster than tall ones do not preclude the possibility that other factors may be much more important within any particular clade [30]. In addition, it is possible that the size of the dataset assembled here (204 spp.) is insufficient to recover a significant effect of plant

height amidst the many other influences on diversification rate. The diversification history of Alooideae appears to have been dominated by two instances of increased diversification rate. The first one happened within the genus *Haworthia* which contains some of the shortest Alooideae, an observation that actually supports a role for reduced plant height in accelerating diversification. However, the most drastic diversification rate increase occurred within the genus *Aloe* in a clade of predominantly tall plants (Fig 2). While the exact reasons for this diversification rate shift remain to be studied, it is interesting to note that *Aloe* is the only genus of Alooideae that has managed to disperse outside of southern Africa, having colonized most of sub-Saharan Africa, Madagascar, and the Arabian Peninsula [41]. This extensive dispersal across Africa has provided *Aloe* species with ample ecological opportunity, possibly stimulating diversification rates to a greater extent than a decrease in plant height would have done.

## Clades of short plants pack more species per unit distance than others

While height does not seem to strongly influence diversification rate across Alooideae, we did find a strong negative effect of plant height on diversification density. This lends support to the hypothesis that, within a set geographic area, smaller-bodied plant species are both more likely to speciate and less likely to go extinct than larger-bodied plants. Increased speciation densities might arise both because gene flow across the area might be more easily interrupted and because local adaptation to varied habitats present within the region might be more frequent for small compared to large plants [20,30]. Such a mechanism could obviously lead to higher speciation rates per unit time in small plants but more directly predicts a higher probability of speciation per unit distance, regardless of the waiting time for speciation to happen or the duration of the speciation process. On the other hand, reduced extinction densities in small plants should arise because, in comparison to taller plants occupying the same geographic area, small plants will benefit from both increased population size [36] and decreased competition with other plants due to higher niche differentiation [35].

Our diversification density metric is most useful as a measure of the spatial scale of diversification when lineages/species have undergone little or no post-speciational range movement or range contraction [83]. The Cape flora of South Africa is an excellent example, the historical stability of the Cape environment fostering the evolution of biota typified by low dispersability and extreme range-restrictedness [84–86]. In such a system, at least, our diversification density metric provides a useful basis for approximating the spatial scale of diversification and drawing comparisons between lineages. This also appears to be the case in Alooideae, where the level of post-speciational range movement has been insufficient to erase broad clade-specific differences in diversification density. In contrast, in highly-perturbed systems, such as the postglacial flora of temperate Eurasia [87] or in clades that have undergone frequent long-distance dispersal events [88], the geographic signature of speciation is likely to be eroded rapidly and our metric is likely to be more strongly influenced by the dynamics of range changes (both movements and contraction/expansion) and their influence on extinction. Thus, as with diversification rate, the underlying drivers of diversification density need to be interpreted cautiously. The metric, which ultimately captures the spatial scale of species accumulation through a lineage's history, will broadly be influenced by the dynamic interplay between range change, diversification and coexistence. For example, while high diversification density could imply frequent speciation across small spatial scales, it could also arise through contraction of a lineage's range with no associated extinction or when traits evolve within a lineage that promote coexistence of species during repeated post-speciational range expansions.

## Consequences for patterns of diversity within the southern African flora

Relative to its rather modest area and intermediate latitude, the southern African flora is one of the richest on Earth [89]. Alooideae is among the most emblematic clades of this region and its species occur throughout southern Africa in all major vegetation types: fynbos, grassland, desert and forest. Within the southern African flora, the fynbos vegetation of the Cape Floristic region harbors the greatest number of plant species and is accordingly recognized as a global biodiversity hotspot [4]. Although less species-rich overall than the fynbos flora, the winter-rainfall desert flora of southern Africa (*i.e.*, the Succulent Karoo flora) has unusually high levels of species richness and endemism per unit area [90,91]. Interestingly, the Succulent Karoo flora harbors exceptionally high numbers of miniature succulents [90,92]. Our results demonstrating that plant height correlates negatively with diversification density in Alooideae suggest an explanation for the high levels of diversity per unit area found in the Succulent Karoo: these dry environments might select for short plant statures, with knock-on consequences for diversification density in particular plant clades, together contributing high species richness over relatively small spatial scales. Further studies of the influence of plant height on diversification density in other important clades of the southern African flora that are well represented in the Succulent Karoo and whose species show extensive height variation are needed to test the generality of this explanation. Especially suitable candidates for tackling this question include tribe Ruschioideae (Aizoaceae) and the genera *Cotyledon* (Crassulaceae), *Crassula* (Crassulaceae) and *Pelargonium* (Geraniaceae).

## Conclusion

In this study we show that the preponderance of short-stature species in Alooideae is not a primary consequence of accelerated diversification in lineages of small plants, but rather due to a general tendency for species to evolve towards smaller heights. We show that, while plant size has not strongly influenced diversification rate (*i.e.* diversification per unit time) in Alooideae, clades of short-stature species achieve significantly higher diversification densities than clades of tall-stature species. Our results thus demonstrate that diversification rate and density can be decoupled. We suggest there is a need to expand the focus of macroevolutionary studies to adopt a wider concept of diversification, one which includes not only the temporal aspect of diversification, but also explicitly considers its spatial component.

## Supporting information

**S1 Appendix. Phylogenetic inference.**
(DOC)

**S2 Appendix. Measuring diversification density.**
(DOC)

**S1 Fig.**
(TIF)

## Author Contributions

**Conceptualization:** Florian C. Boucher, Anne-Sophie Quatela, Allan G. Ellis, G. Anthony Verboom.

**Data curation:** Florian C. Boucher, Anne-Sophie Quatela.

**Methodology:** Florian C. Boucher.

**Writing – original draft:** Florian C. Boucher.

**Writing – review & editing:** Anne-Sophie Quatela, Allan G. Ellis, G. Anthony Verboom.

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
