## [Decision Letter · Decision Letter 0]

14 Nov 2019

PONE-D-19-21695

DIVERSIFICATION RATE VS. DIVERSIFICATION DENSITY: DECOUPLED CONSEQUENCES OF PLANT HEIGHT FOR DIVERSIFICATION OF ALOOIDEAE IN TIME AND SPACE

PLOS ONE

Dear Dr. Boucher,

Thank you for submitting your manuscript to PLOS ONE. After careful consideration, we feel that it has merit but does not fully meet PLOS ONE’s publication criteria as it currently stands. Therefore, we invite you to submit a revised version of the manuscript that addresses the points raised during the review process.

Dear authors, I can finally return your paper. I agree with reviewers that you present an intriguing and exciting study on the dynamics of diversification in the Alooideae tribe. Although some points needs to be considered. Looking foward for the next version of your manuscript.

We would appreciate receiving your revised manuscript by Dec 29 2019 11:59PM. To enhance the reproducibility of your results, we recommend that if applicable you deposit your laboratory protocols in protocols.io, where a protocol can be assigned its own identifier (DOI) such that it can be cited independently in the future. For instructions see: http://journals.plos.org/plosone/s/submission-guidelines#loc-laboratory-protocols

We look forward to receiving your revised manuscript.

Kind regards,

Juliana Hipólito, Phd

Academic Editor

PLOS ONE

Journal Requirements:

Additional Editor Comments (if provided):

Reviewers' comments:

Reviewer's Responses to Questions

**Comments to the Author**

1. Is the manuscript technically sound, and do the data support the conclusions?

Reviewer #1: Yes

Reviewer #2: Partly

2. Has the statistical analysis been performed appropriately and rigorously? 

Reviewer #1: Yes

Reviewer #2: No

3. Have the authors made all data underlying the findings in their manuscript fully available?

Reviewer #1: Yes

Reviewer #2: Yes

4. Is the manuscript presented in an intelligible fashion and written in standard English?

Reviewer #1: Yes

Reviewer #2: Yes

5. Review Comments to the Author

Reviewer #1: A generally interesting and well done paper that attempts to advance methods in estimating diversification rates by introducing the idea of density. The addition of density is a novel way of interpreting the macroevolutionary scenario and I find it generally to be a useful topic. However, the text neglects to consider the effect of biogeographic history on density. Long distance dispersal in particular can result in situation that decouple density and net diversification rate. For example, when species from the dense area that exhibits high net div disperse to oceanic islands or other low-density areas, a decoupled pattern occurs. This might not be realistic for the study group detailed here, but it is realistic for a lot of other organisms and should be discussed in order to make the topic more broadly relevant.

The elephant in the room, which is not addressed here, and in fact not addressed in many papers attempting to find generally patterns that explain net div is the question of whether the dataset is sufficient to answer the question. The phylogeny used has one clade with high net div, and a couple of clades wth moderately elevated rates. The one clade with a high rate is also the clade with the shortest plants. So, in effect, there are not a lot of opportunities or ‘data points’ that allow for the discovery of a general pattern.

The scholarship is generally excellent but neglects to cite a number of papers addressing size evolution in plants. Few researchers have attempted to address this topic, so it seems particularly relevant and easy to include these. I have listed a few papers analyzing molecular evolution and diversification rates along with plant size below.

Molecular evolution and plant size:

Smith, S.A. and Donoghue, M.J., 2008. Rates of molecular evolution are linked to life history in flowering plants. science, 322(5898), pp.86-89.

Lanfear, R., Ho, S.Y., Davies, T.J., Moles, A.T., Aarssen, L., Swenson, N.G., Warman, L., Zanne, A.E. and Allen, A.P., 2013. Taller plants have lower rates of molecular evolution. Nature Communications, 4, p.1879.

Diversification rate and plant size:

Sundue, M.A., Testo, W.L. and Ranker, T.A., 2015. Morphological innovation, ecological opportunity, and the radiation of a major vascular epiphyte lineage. Evolution, 69(9), pp.2482-2495.

Ramírez-Barahona, S., Barrera-Redondo, J. and Eguiarte, L.E., 2016. Rates of ecological divergence and body size evolution are correlated with species diversification in scaly tree ferns. Proceedings of the Royal Society B: Biological Sciences, 283(1834), p.20161098.

Testo, W.L. and Sundue, M.A., 2018. Are rates of species diversification and body size evolution coupled in the ferns?. American journal of botany, 105(3), pp.525-535.

Reviewer #2: The authors present an intriguing and exciting study on the dynamics of diversification in the Alooideae tribe. They formalize the concept of ‘diversification density,’ a spatial counterpart to diversification rate, which may be a useful metric for understanding how lineages assemble into communities or how biodiversity hotspots form. The authors then integrate this new metric with plant height to test the hypotheses that shorter statured species should diversify more rapidly and across smaller spatial distances. They find support for the latter hypothesis only. This manuscript was overall a pleasure to read and I believe it provides an important advancement on our understanding of speciation dynamics across time and space.

Before raising some points that I believe need to be addressed, I wanted to offer a quick apology to the authors for being tardy with my review. It’s taken me a whole month to provide my comments to the editors. The holdup is my fault, not theirs. Sorry!

My primary concerns with this manuscript relate to the methodological. Some of the analyses conducted are very rigorous, cutting edge, and elegant. Others, such as the molecular dating and particularly the treatment of species distributions was cursory and possibly inappropriate.

Regarding the molecular dating, the authors used a single secondary calibration and a deprecated penalized likelihood method. Given the rigor of the ancestral reconstructions and STRAPP analyses, I was surprised that the dating itself was not given more attention. Though the authors suggest that absolute dates are less important in this study because they are concerned with relative diversification rates, this assumes that clade ages are dependent only on the root age. Other, newer PL methods (e.g., chronos, treePL) offer more rigorous data-exploration and cross-validation approaches.

I have several concerns regarding the species distribution data. First, the authors downloaded species distributions from the SANBI database and apparently proceeded with no further quality control. Species distribution databases – even very good ones – have many errors. Because species ranges are so important to this study, erroneous records are a problem. Second, there is tremendous bias in species collections – both geographically and taxonomically. What steps were taken to ensure evenness of sampling across species and areas? If there is unevenness in collections, then the measure of species area as the sum of QDS is going to be incomparable across species and a different method (convex hull area?) could be more appropriate. Third, the authors restricted their analyses to contiguous Southern Africa. By doing so, they are artificially and unevenly reducing species range sizes (and inflating density) for those species (lineages) that extend beyond contiguous southern Africa.

Minor Comments

I found the framing of this study in the context of biodiversity hotspots to be somewhat confusing. I don’t have a specific request for how it can be approved, but rather simply want to explain why I was confused in case the authors decide some edits would help. Based on the introduction (and abstract) as written, I was expecting analyses about net diversity within areas across lineages (i.e., area-centric analyses), but instead the analyses are about species density within lineages across areas (i.e., lineage-centric analyses).. An area-centric analysis would ask, e.g., does the species richness in area of a given size (density) increase as the average species height within that area decreases? Whereas, this study (as I understood it) asks, do shorter statured species diversify more rapidly and across smaller spatial scales than taller species? A number of metrics already exist for the area-centric approach (e.g., those coming out of the Mishler and Laffan labs), so I do appreciate the formalization the authors are providing for the lineage-centric perspective. And, of course diversification density is relevant to the formation of biodiversity hotspots. Fourth, the implication of the approach is that the area currently occupied by species is the same area in which the species originated.

Line 25: consider rephrasing “has influenced” to “is associated with”

Paragraph beginning line 56: This is a fairly cursory literature review.

Line 65: Even greater than area occupied is the extent to which that area is partitioned by individual species, at least in sedges

Line 70: But this is context dependent. Widespread boreal lineages are still species poor.

Line 84: Not necessarily. Species with reduced dispersal distances tend to have higher diversification rates (Givnish 2010)

Line 86: Not necessarily, would depend on how clades are defined.

Line 135: Thanks for including this. Scripts were well annotated and easy to follow.

Study Group section: please also include information on dispersal syndrome (which appears to be very important based on the plant height papers cited). Does dispersal syndrome vary across the study group?

Diversification Density metric: Perhaps this is outside the scope of the study…but it seems that lineage density would be more immediately relevant than species density. Besides the subjectivity and variability of species definitions, either a measure of Faith’s phylogenetic diversity or mean phylogenetic distance would more naturally include the phylogeny than the time slice approach, and also better accommodate incomplete taxon sampling.

Line 336. If these results were unexpected, and previous studies with different results were credible, perhaps Kumara should be constrained as monophyletic.

Line 482: they also on average have shorter generation times as they sooner to reproduce than species that first require vertical growth.

6. PLOS authors have the option to publish the peer review history of their article (what does this mean?). If published, this will include your full peer review and any attached files.

Reviewer #1: No

Reviewer #2: No

---

## [Author Response · Author response to Decision Letter 0]

25 Feb 2020

Please see the attached file for a point by point response to the reviewers.

---

## [Decision Letter · Decision Letter 1]

11 May 2020

DIVERSIFICATION RATE VS. DIVERSIFICATION DENSITY: DECOUPLED CONSEQUENCES OF PLANT HEIGHT FOR DIVERSIFICATION OF ALOOIDEAE IN TIME AND SPACE

PONE-D-19-21695R1

Dear Dr. Boucher,

We are pleased to inform you that your manuscript has been judged scientifically suitable for publication and will be formally accepted for publication once it complies with all outstanding technical requirements.

With kind regards,

Juliana Hipólito, Phd

Academic Editor

PLOS ONE

Additional Editor Comments (optional):

Dear author's, I apologize for the long delay in finally giving an answer on the manuscript. due to the pandemic and the difficulty in finding available reviewers, the process has become increasingly difficult. At least now I can give you a positive answer.

Reviewers' comments:

Reviewer's Responses to Questions

**Comments to the Author**

1. If the authors have adequately addressed your comments raised in a previous round of review and you feel that this manuscript is now acceptable for publication, you may indicate that here to bypass the “Comments to the Author” section, enter your conflict of interest statement in the “Confidential to Editor” section, and submit your "Accept" recommendation.

Reviewer #2: All comments have been addressed

Reviewer #3: All comments have been addressed

2. Is the manuscript technically sound, and do the data support the conclusions?

Reviewer #2: Yes

Reviewer #3: Yes

3. Has the statistical analysis been performed appropriately and rigorously? 

Reviewer #2: Yes

Reviewer #3: Yes

4. Have the authors made all data underlying the findings in their manuscript fully available?

Reviewer #2: Yes

Reviewer #3: Yes

5. Is the manuscript presented in an intelligible fashion and written in standard English?

Reviewer #2: Yes

Reviewer #3: Yes

6. Review Comments to the Author

Reviewer #2: The authors have sufficiently addressed all questions that I raised in my initial review. I think that this is a great manuscript and presents useful data and tools that will be widely appreciated.

Reviewer #3: All the concerns of the referees have been addressed,so I suggest that the revised version be accepted for publication

7. PLOS authors have the option to publish the peer review history of their article (what does this mean?). If published, this will include your full peer review and any attached files.

Reviewer #2: No

Reviewer #3: No

---

## [Editor Report · Acceptance letter]

14 May 2020

PONE-D-19-21695R1 

DIVERSIFICATION RATE VS. DIVERSIFICATION DENSITY: DECOUPLED CONSEQUENCES OF PLANT HEIGHT FOR DIVERSIFICATION OF ALOOIDEAE IN TIME AND SPACE 

Dear Dr. Boucher:

I am pleased to inform you that your manuscript has been deemed suitable for publication in PLOS ONE. Congratulations! Your manuscript is now with our production department. 

With kind regards,

on behalf of

Dr. Juliana Hipólito 

Academic Editor

PLOS ONE